# CogVLN: Cognitive Map-Guided Vision-and-Language Navigation in Large-Scale Environments

## Abstract

Vision-and-Language Navigation (VLN) requires agents to interpret language instructions and navigate to target locations via visual observations. While progress has been made indoors, large-scale outdoor VLN remains underexplored. The large-scale environment representation as a primary challenge. Though dense maps (e.g., point clouds) enable flexible environment modeling, their high memory usage fails to meet navigation's real-time needs. Additionally, aligning long language instructions with complex environments remains a notable issue. In this paper, we introduce CogVLN, a novel method for VLN in large-scale environments. First, inspired by how humans encode environments, we propose a method for constructing a cognitive map to represent large-scale environments. This method prioritizes encoding key scenes that embody distinct environmental features, while allocating fewer coding resources to scenes with higher consistency. Subsequently, leveraging the constructed cognitive map, we design three core functional modules: a localization module responsible for identifying start and goal vertex, a path planning module tasked with planning navigation routes, and a navigation module dedicated to carrying out the navigation task. During the navigation process, driven by the multimodal large language model (MLLM), CogVLN expresses scene information and receive user feedback in an interactive manner, and further dynamically adjusts the route accordingly. Experimental results in the CARLA Town01 and Town07 environments demonstrate the remarkable performance of our CogVLN.

## 1 Introduction

Vision-and-Language Navigation (VLN) requires the agent to complete navigation task by following human instructions in 3D environments (Anderson et al., 2018b; Gu et al., 2022), which is a complex task combining techniques from computer vision, natural language processing and robot field. As an important research direction in embodied intelligence, VLN demonstrates broad application potential in real-world scenarios, such as indoor smart home robots, outdoor delivery robots, and logistics systems. Most existing methods primarily focus on indoor environments and have achieved strong navigation performance in both discrete and continuous settings. (Wang et al., 2019; Chen et al., 2021; Wang et al., 2023a; Chen et al., 2025). However, when VLN tasks are extended to large-scale outdoor environments, these methods often struggle to generalize due to increased scene complexity.

In large-scale environments, building an efficient spatial representation is a prerequisite for navigation. Dense maps can clearly reproduce the structure of the environment. For example, point clouds flexibly capture arbitrary 3D structure by aggregating 3D points with their local descriptors (Shan et al., 2020; Zhang et al., 2014). However, achieving fast retrieval typically requires additional index structures (e.g., KD-trees), which incur substantial storage and time overhead. Local point clouds and sparse surface maps reduce storage and computation, but there are still significant difficulties in applying it to real-time navigation. (Campos et al., 2021; Shan et al., 2020). Topological maps abstract the environment into semantic vertex and connected edges, compressing large spaces into a structure that is suitable for reasoning and planning, naturally applicable to the representation of large-scale environments (Zemskova & Yudin, 2024). Both Touchdown (Chen et al., 2019) and

Map2Seq (Schumann & Riezler, 2020) datasets represent city environments with topological graphs, and provide benchmark datasets for outdoor VLN task (Xiang et al., 2020; Tian et al., 2024). But, they collected data at equal intervals for all the vertex, which can lead to redundant or sparse environmental representations. Furthermore, for the existing outdoor navigation methods (Huang et al., 2025; He et al., 2025), agents mainly rely on a single set of language instructions provided at the start of navigation to complete the task, and their success deeply depends on the degree of alignment between these instructions and visual observations. However, in large-scale environments, both VLN instructions and agent trajectories are much longer than in indoor settings. The increased complexity of the trajectories and the length of the instructions make crossmodal alignment more difficult, thereby limiting navigation performance.

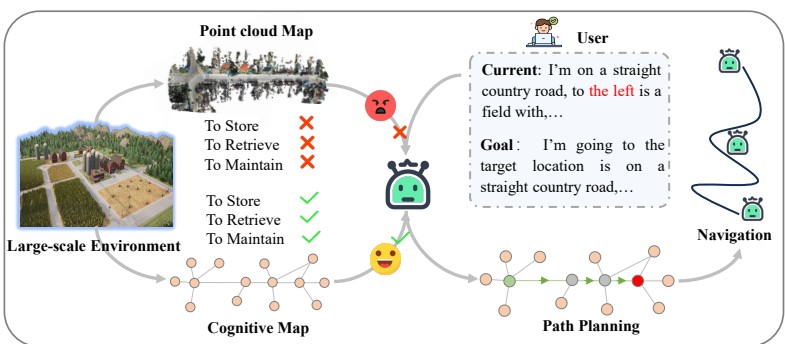

Figure 1: Overview of the CogVLN framework.

In this work, we introduce the CogVLN, a vision-and-language navigation framework for large-scale environment which is shown in Figure 1. Cognitive maps describe the internal spatial representations in human and animal brains. Humans and animals demonstrate remarkable navigational abilities in complex environments through the use of cognitive maps. Inspired by this, we propose a method for constructing cognitive map to represent the large-scale environment. The method leverages visual and semantic cues to remove redundant scenes via similarity computation and preserve distinctive ones, representing the large-scale environment in a topological form. In addition, based on cognitive map, we propose a localization module to locate the start and goal positions, a path planning module to plan navigation routes, and a navigation module to complete the navigation task. Unlike existing methods, we do not input navigation instructions all at once. Instead, CogVLN progressively outputs scene information throughout the entire navigation process, adjusting the route in real-time based on user feedback to achieve interactive navigation in large-scale environments, hereby avoiding the complexity of cross-modal alignment, hereby avoiding the complexity of cross-modal alignment.

In summary, our contributions can be summarized as follows:

- We propose a method for constructing cognitive maps to represent large-scale environments. The cognitive maps are not only more compact than dense representations like point clouds but also more reasonable and efficient than conventional topological maps.
- Based on the cognitive map, we introduce localization, path planning, and navigation modules that with the help of the understanding capabilities of Multimodal Large Language Models (MLLMs) to enable interactive navigation.
- We validated our method in two challenging large-scale environments, Town01 and Town07 in CARLA. Without any training or fine-tuning, our proposed CogVLN achieved excellent performance, with navigation success rates of 34.17% and 28.33% respectively.

## 2 RELATED WORK

### 2.1 INDOOR VISION-AND-LANGUAGE NAVIGATION

In the field of VLN, early studies primarily focused on discretized environments, which are represented as fixed graph structures, where agents leverage visual and language inputs to move be-

tween predefined nodes (Hong et al., 2021; Wang et al., 2023b). However, this strategy constrains its applicability in complex real-world environments. More recent studies have shifted their focus toward navigation tasks in continuous environments (Krantz et al., 2020; An et al., 2024; Chen et al., 2025), where agents perform primitive actions to achieve more realistic navigation through low-level control or waypoint-based methods (Hong et al., 2022). Recent advancements in Multi-modal Large Language Models (MLLMs) have opened up new possibilities for VLN. For example, MapGPT (Chen et al., 2024) uses GPT-4V to perceive environmental information and encodes it into text for multi-step prediction. OpenNav (Qiao et al., 2025) specifically explores the potential of open-source LLMs in zero-shot VLN tasks. NavGPT (Zhou et al., 2024) employs GPT as the core of the navigation system, not only for visual observation and history recording but also for inference and decision-making. In our work, we utilize the understanding capabilities of the MLLMs for scenes and instructions, and gradually output navigation information to achieve interactive navigation with the users..

## 2.2 VLN in Large-scale Environment

In large-scale environments, the VLN task becomes more complex, and traditional indoor VLN methods are not directly applicable. Therefore, researchers have proposed various VLN methods for outdoor scenarios. Li et al. (Li et al., 2024) started from videos and proposed the VLN-VIDEO, they generate navigation instructions through template filling and predict actions using image rotation similarity, constructing large-scale training data from driving videos to enhance the navigation capabilities of agent in outdoor Scenes. Vasudevan et al. (Vasudevan et al., 2021) addressed the problem of insufficient fine-grained data alignment by constructing a segmentally aligned landmark–direction dataset and proposed a model that combines dual attention with explicit spatial memory, thereby achieving efficient city-level navigation. Tian et al. (Tian et al., 2024) drew on human navigation experience and emphasized the key role of spatial localization in outdoor VLN. Schumann et al. (Schumann et al., 2024) proposed transforming navigation tasks entirely into text sequences and appending environmental observations as textual descriptions to the prompts, thereby enabling navigation under few-shot learning conditions. And MMCNav (Zhang et al., 2025) is the first to tackle the VLN task with a multi-agent framework, where each agent assumes a specific role within the system and collaborates to accomplish its assigned tasks, thereby enhancing outdoor navigation capabilities. Different from the above, we construct a brain-inspired cognitive map to represent the large-scale environment, and based on the cognitive map to proceed the localization, path planning, and navigation.

## 2.3 Topological Representation in Navigation

In large-scale outdoor environments, effective spatial representation is a core requirement of VLN. Dense maps such as point clouds can flexibly represent the environment, but their storage requirements and retrieval speeds limit their application in real-time navigation. (Wang et al., 2024). In contrast, topological graphs, which abstract the representation of the environment, are more suitable for environment representation. ETPNav (An et al., 2024) represents the environment as a dynamic topological graph and continuously evolves and updates it during navigation. Mem4Nav (He et al., 2025) employs topological representations to capture high-level environmental semantics, such as landmarks and intersections, as well as their spatial connectivity, enabling structured reasoning for navigation. MapGPT (Chen et al., 2024) leverages topological representations to explicitly encode environmental structures into prompts, serving as the basis for LLM-based navigation decisions. As a type of topological map, cognitive maps are considered internal representations of the environment within the brain during navigation by humans and animals. Inspired by this, in our work, We construct a cognitive map to represent large-scale outdoor scenes, which encodes the environment from both visual and semantic perspectives, eliminating redundant features and retaining key features.

## 3 Cognitive Map-Guided VLN

### 3.1 Overview

We address the VLN task in large-scale environments, where the agent navigates to a destination following natural language instruction. Unlike previous works (Huang et al., 2025; He et al., 2025),

which inputs all instructions at once and converts the VLN task into sequential decision-making and multimodal alignment problems. We propose the CogVLN, a vision-and-language navigation framework. First, we present a method constructing cognitive map to represent the large-scale environments. Then based on the cognitive map, driven by the Multimodal Large Language Model (MLLM), the proposed CogVLN enables dynamic interactions between agents and humans, thereby achieving robust navigation performance in large-scale environments.

In this work, the scene information we utilized is based on the previous work, NeuroBayesS-LAM (Zeng et al., 2020), which integrates vestibular and visual cues through a Bayesian attractor network and employs the neural dynamics of grid and head direction cells to generate a topological map with visual information, connected relation, and coordinate information. In our research, the navigation environment is described as an undirected graph $G = (W, E)$, where $W = \{w_i\}_{i=1...N}$ and $E = \{e_j\}_{j=1...J}$ represent the sets of waypoints and edges, respectively. The waypoints $W = (V, C)$ include visual information $V$ and coordinate information $C$, which can be represented as $V = \{v_i\}_{i=1...N}$ and $C = \{c_i = (x_i, y_i)\}_{i=1...N}$, $N$ is the total number of vertex.

## 3.2 COGNITIVE MAP CONSTRUCTION

Inspired by the human use of cognitive maps to represent environments in navigation, we adopt a cognitive map to represent large-scale environment in navigation tasks. Humans tend to encode multiple cues within a scene, where similar areas are merged into common templates, leading to only a limited number of distinctive details being preserved. Following this principle, we encode waypoint information from both visual and semantic perspectives, filtering out highly similar and redundant scenarios while retaining distinctive ones that align with human cognition and carry representative significance.

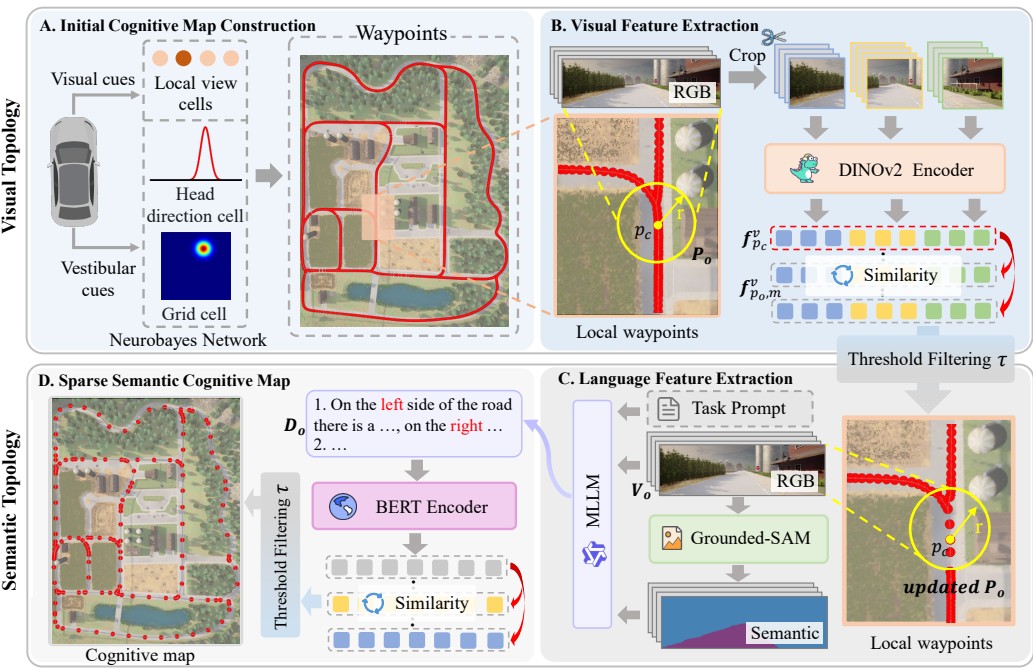

Figure 2: Framework for Cognitive Map Constructing. A involves obtaining the waypoint information through NeuroBayesSLAM. B filters out redundant scenes based on visual similarity. C extracts semantic features of the scene. D constructs the complete cognitive map using semantic similarity.

**Visual Topology.** As shown in Figure 3.2 (A), we obtained the dense waypoints of the environment by integrating visual and vestibular cues through a NeuroBayes Network. Then, in Part B, we define a set $P = \{p_k\}_{k=1...K}$ to store the unprocessed waypoints. Initially, with $K = N$, the set $P$ contains all waypoints. We compute the centroid of all vertex in $P$ and select the vertice $p_c$ closest

to the centroid as the center, based on which we construct the subset $P_o = \{p_{o,m}\}_{m=1...M}$ with radius $r$. For the center vertice $p_c$, its RGB image is evenly cropped into three parts and encoded with DINOv2 (Oquab et al., 2023) to obtain the feature vectors $f_{p_c}^{v(1)}$, $f_{p_c}^{v(2)}$, and $f_{p_c}^{v(3)}$, which are concatenated to form the visual representation $f_{p_c}^v$. The same procedure is applied to all other vertex in the subset $P_o$, yielding their visual representations $f_{p_{o,m}(m=1...M-1)}^v$. Having obtained the visual features $f_{p_{o,m}}$, we employ cosine similarity to identify redundant features that should not be encoded in the cognitive map.

$$s_m^v = \frac{(f_{p_c}^v)^\top f_{p_{o,m}}^v}{\left\| f_{p_c}^v \right\| \left\| f_{p_{o,m}}^v \right\|}, m = 1, ..., M - 1. \tag{1}$$

where $s_m^v$ denotes the visual similarity score between the $m$-th vertice and the center vertice. Based on the threshold $\tau$, we obtain a set $P_o^v = \{\{p_{o,m}\}_{m \in \{1,...,M-1\}} | s_m^v \geq \tau\}$, which contains the vertex that are not specially encoded or memorized in the construction of the cognitive map. Therefore, all connections (edges) of these vertex are assigned to the center vertice $p_c$, while the vertex themselves are removed from the set subset $P_o$. Assuming that a total of $L$ vertex are removed, the set $P_o$ is updated as $P_o = \{p_{o,m}\}_{m=1...M-L}$.

**Semantic Topology.** Semantic understanding of a scene involves higher-level reasoning and requires certain prior knowledge to be properly interpreted. Therefore, we employ a pretrained Multimodal Large Language Model (MLLM) to perceive the semantic information of the scene. As shown in Figure 3.2 (C), we first feed the RGB images of the vertex in the updated set $P_o$ into the Grounded-SAM (Ren et al., 2024) model to obtain road semantic maps under the guidance of the [road] semantic label. The RGB images, the corresponding road semantic maps, and the prompt are input into the MLLM to generate spatial descriptions $D_o$ centered on the road. Subsequently, the BERT (Devlin et al., 2019) model is employed to encode the descriptions, generating the semantic vectors $f_{p_{o,m}(m=1...M-L)}^s$. The above steps can be defined as follows:

$$D_o = MLLM((V_o, GroundedSAM(V_o), Prompt)) \tag{2}$$

$$f_{p_{o,m}}^s = BERT(D_o) \tag{3}$$

where $V_o$ denotes the visual information derived from the updated set $P_o$. Similar to the visual topology, we compute the semantic cosine similarity between the center vertice $p_c$ and the other vertex, thereby obtaining a set $P_o^s = \{\{p_{o,m}\}_{m \in \{1,...,M-L-1\}} | s_m^s \geq \tau\}$. The connections (edges) of the vertex in set $P_o^s$ are reassigned to the center vertice $p_c$, while the vertex themselves are removed from $P_o$. The set $P$ is synchronously updated by deleting all vertex that were eliminated in the two rounds.

Based on the updated set $P$, the above process is repeated until $P$ becomes empty. In this way, a cognitive map $G_{cog}$ for navigation is constructed, which aligns with human scene cognition by retaining only key information about the environment while eliminating redundant details.

## 3.3 MULTI-MODAL LOCALIZATION, PLANNING AND NAVIGATION

Based on cognitive map , CogVLN will utilize multimodal information to perform the entire navigation task. We designed three modules in total: the localization module, the path planning module, and the navigation module. Their detailed structure and functionality are illustrated in 3.3.

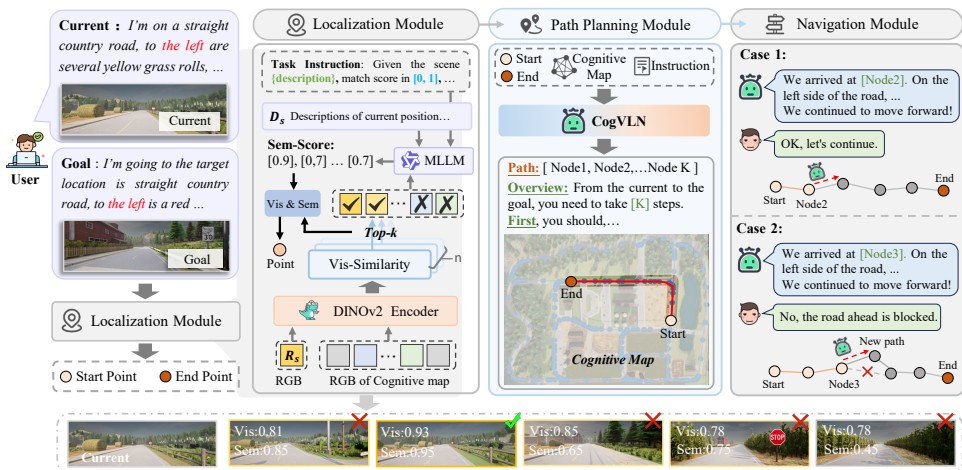

Figure 3: Framework for Localization, Path Planning and Navigation. The agent receives description information about the start and goal locations, along with RGB images. The localization module identifies the positions of the start and goal. The path planning module is responsible for generating the route. The navigation module then executes the navigation step by step in an interactive manner.

**Localization Module**    In CogVLN, the localization processes for the start and goal vertex are identical. Below, we take the localization of the start vertice as an example for illustration. First, the localization module receives the language description $D_s$ and RGB image $R_s$ of the start vertice. For the RGB image, the module encodes it using DINOv2 (Oquab et al., 2023), and also encodes the RGB images of all vertex in the cognitive map $G_{cog}$. It then computes the similarity between the input image and each map vertex, and selects the top-5 vertex with the highest similarity scores. Subsequently, the RGB images of the top-5 candidate vertex, along with the given language description $D_s$ of the start vertice, are fed into a Multimodal Large Language Model (MLLM). Guided by a task instruction prompt, the MLLM leverages its strong multimodal understanding capabilities to perform semantic matching between the described scene and each of the five candidate images. The output is a semantic similarity score (sem-score) for each of the five images. Finally, the start vertice is determined by computing a weighted score that combines both visual similarity and semantic similarity, selecting the vertex with the highest overall score as the start location. Then repeat the above process to obtain the goal location.

**Path Planning Module**    After localizing the start and goal vertex of the navigation task, we introduce a path planning module to path the route. Specifically, we feed the constructed cognitive map $G_{cog}$, the start and goal IDs, and the task prompt into CogVLN. Guided by the prompt and assisted by an MLLM, CogVLN develops a thorough understanding of the cognitive map and plans an optimal route. The output is a sequence of vertice IDs, such as Path: [Vertice1, Vertice2 . . . ]. Within the path planning module, the agent not only performs global planning with the MLLM but also produces step-by-step descriptions of the navigation process, enabling users to gain an initial understanding of the entire route.

**Navigation Module**    Finally, we design a navigation module to execute the navigation task. Starting from the star vertice, the agent passes through the planned waypoints step by step. Upon arriving at each new waypoint, the agent first performs image matching between the RGB image stored in the cognitive map $G_{cog}$ and the image captured by its current camera to verify the accuracy of the planned route. Then, the agent outputs a description of the current scene, and according to the user's feedback to further confirm the next navigation step. If it receives an [OK] response, the agent continues along the planned route to the next waypoint (Corresponding to case 1). If a [NO] response is received, the agent recognizes that there may be an issue ahead and re-plans the route to avoid the problematic waypoint (Corresponding to case 2). This interactive navigation process avoids the alignment difficulty caused by long instructions and long distances, and enables dynamic path adjustment based on user feedback.

# 4 EXPERIMENTS

## 4.1 EXPERIMENTAL SETUP

**Simulation Platform and Datasets.** The experimental environment in this study is based on CARLA, an open-source simulation platform widely used in autonomous driving research. We conducted experiments in Town01 which features a basic urban layout, and in Town07 which a rural setting with diverse road structures. We used NeuroBayesSLAM (Zeng et al., 2020) algorithm to collect the datasets in Town01 and Town07. We obtained 2555 and 2268 waypoints in Town01 and Town07, respectively. For each waypoint, we recorded the coordinate information and an RGB image with resolution of $1242 \times 375$, and we also captured the links between adjacent waypoints. Each environment set have 120 different navigation tasks with different start-goal pairs, each start-goal pair is randomly chosen. All algorithms were run on a unified Ubuntu 22.04 platform equipped with an Intel® Xeon® Gold 6230 (2.10 GHz) CPU and an NVIDIA RTX 4090 GPU with 24 GB VRAM, ensuring consistent computational performance across all experiments.

**Evaluation Metrics.** We adapt evaluation metrics commonly used in VLN task (Anderson et al., 2018a), they are success rate (SR), oracle success rate (OSR), success weighted by path length (SPL), and navigation error (NE). SR measures the proportion of successful in which the agent reaches the goal. A navigation attempt is deemed successful only if the agent correctly localizes the start and goal on the cognitive map, all visited waypoints correspond to vertex in the map, and the traversed edges respect the map's connectivity. SPL captures both success and path efficiency, favoring near-optimal routes. We use A* Hart et al. (1968) to define the reference shortest path between each start–goal pair. OSR evaluates whether the agent reaches a specific location on the optimal route, even if it fails to arrive at the final destination. We deem OSR satisfied when the agent reaches a vertex that is directly connected (adjacent) to the goal. NE measures the average distance between the agent's final position and the goal.

## 4.2 COMPARISON BASELINE.

Table 1 presents the performance of our method in the Town01 and Town07 environments, where we adopt Qwen3-VL-Plus as the built-in MLLM for CogVLN. It is evident that CogVLN achieves strong performance without any training or fine-tuning, reaching success rates (SR) of 34.17% in Town01 and 28.33% in Town07.

We reviewed previous VLN work in large-scale environment. Our CogVLN was the closest to (Zeng et al., 2024), but there were still significant differences. PReP fine-tuned the built-in driver of MLLM, which requires a lot of time to build the dataset and train, limiting the scalability of PReP. Moreover, PReP is not a continuous navigation, it needs to stop to think before making the next decision. Therefore, we include only a random policy as a reference, in which the agent uniformly selects an outgoing connection from the current veretice at each step.

Furthermore, we have visualized the navigation process which is shown in Figure 4. We can clearly see that CogVLN is capable of planning the path from Star to End. During the navigation process, descriptions about each vertex will be output. After receiving the [OK] instruction, CogVLN will continue to move forward along the route, such as **Navigation 1** and **Navigation 2**. When CogVLN receives the user's [NO] instruction, it will re-plan the route to reach END and continue the navigation, such as **Navigation 3** and **Navigation 4**.

Table 1: Results on our method compared with the random method.

| Method | Town01 | | | | Town07 | | | |
|--------|------|------|------|------|------|------|------|------|
| | NE↓ | SR↑ | OSR↑ | SPL↑ | NE↓ | SR↑ | OSR↑ | SPL↑ |
| Random | - | 0.83 | 0.83 | 0.16 | - | 0.00 | 0.00 | 0.00 |
| Ours | **3.40** | **34.17** | **54.17** | **17.78** | **4.44** | **28.33** | **41.67** | **17.13** |

Table 2: The comparison results between cognitive maps and equidistant topological maps.

| Radius | Town01 | | | | Town07 | | | |
|--------|--------|--------|--------|--------|--------|--------|--------|--------|
| | NE↓ | SR↑ | OSR↑ | SPL↑ | NE↓ | SR↑ | OSR↑ | SPL↑ |
| Equ | 4.97 | 23.33 | 35.83 | 13.23 | 4.85 | 25.00 | 33.33 | 14.28 |
| Cog | **3.40** | **34.17** | **54.17** | **17.78** | **4.44** | **28.33** | **41.67** | **17.13** |

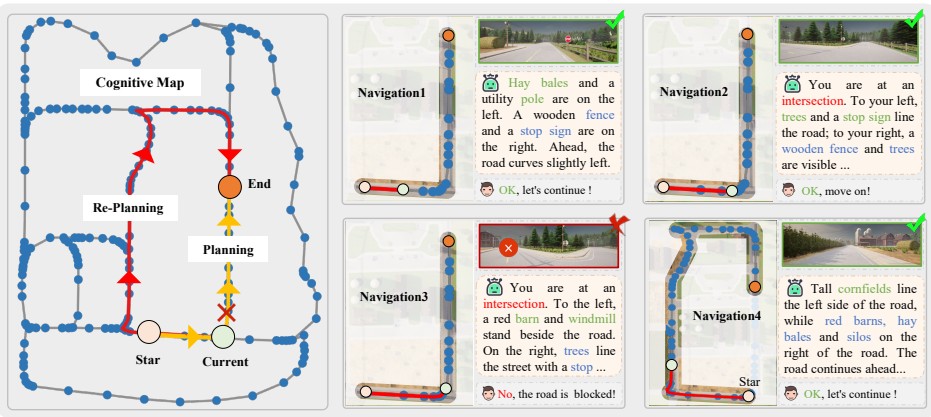

Figure 4: The picture illustrates the process of interactive navigation, including the re-planning of the route through interaction.

## 4.3 ABLATION EXPERIMENTS

**Effectiveness of Cognitive map.** To validate the effectiveness of the proposed cognitive map, we compared it with equidistantly constructed topological graphs (Equ). As shown in Table 3, the proposed cognitive map clearly outperforms the equidistant topological graphs.

In addition, Figure 5 visualizes the cognitive maps under different radius settings, where different colors represent varying waypoint densities. The first row (a) to (d) shows the visualizations of the Town07 scene with radii set to 15, 25, 35, and 45, respectively. As the radius increases, the bottom and right parts of the scene become increasingly sparse, indicating that the waypoints in these areas carry highly similar visual information and are therefore not redundantly encoded. In contrast, the density in the central intersection area remains relatively unchanged, suggesting that the cognitive map preserves representative and informative locations regardless of the increasing topological radius. The second row (e) to (h) presents the corresponding visualizations for the Town01 scene under different radius settings. Similarly, we observe that only the straight boundary roads are gradually pruned, while key features such as turns and intersections are well preserved in the map structure.

We also show the number of vertex retained in the cognitive map when constructed with different radii, as shown in Appendix. It is evident that as the radius increases, the number of vertex first decreases and then gradually stabilizes, indicating that the key feature points of the environment are effectively preserved.

**Effectiveness of Interaction Navigation.** We randomly removed 12 connecting edges on the cognitive map to simulate the situation where the roads are blocked. Without interactive navigation, the different routes would directly lead to navigation failure. However, through interactive navigation, the agent can re-plan the path and navigate to the target location. The navigation results of the re-planned path are shown in Table 3.

**Comparing different MLLMs.** Multimodal large language models (MLLMs) serve as the core driving force of our approach. They are not only responsible for understanding the semantics embed-

Table 3: The experimental results of randomly deleting adjacent edges.

| Radius | Town01 | | | Town07 | | |
|---|---|---|---|---|---|---|
| | SR↑ | OSR↑ | SPL↑ | SR↑ | OSR↑ | SPL↑ |
| Remove | 24.17 | 35.83 | 13.46 | 18.33 | 29.17 | 14.83 |
| Cog Map | **34.17** | **54.17** | **17.78** | **28.33** | **41.67** | **17.13** |

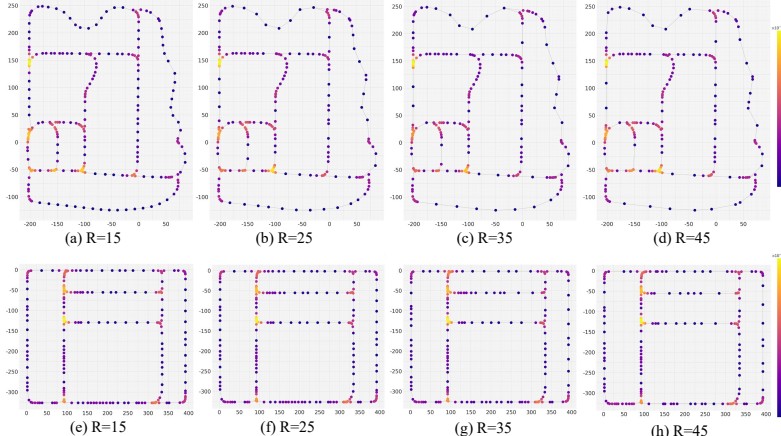

Figure 5: The number of vertex in the cognitive maps constructed with different radii in Town01 and Town07.

ded in RGB images, but also for interpreting the cognitive map and planning navigation paths based on it. Additionally, MLLMs act as the medium for interaction between the user and the agent. We use different MLLMs to perform CogVLN, including DeepSeek-V3.1, GLM-4.5V and Qwen3-VL. Table 4 presents their navigation performance.

Table 4: Results on different Multimodal Large Language Models (MLLMs) in our algorithm.

| Model | Town01 | | | | Town07 | | | |
|---|---|---|---|---|---|---|---|---|
| | NE↓ | SR↑ | OSR↑ | SPL↑ | NE↓ | SR↑ | OSR↑ | SPL↑ |
| Deepseek-V3 | 3.96 | 29.17 | 43.33 | 15.22 | 5.28 | 25.00 | 35.83 | 13.37 |
| Gemini-2.5 | 4.36 | 32.50 | 45.83 | 18.85 | 4.65 | 33.33 | 38.33 | 18.43 |
| Qwen3-VL | 3.40 | 34.17 | 54.17 | 17.78 | 4.44 | 28.33 | 41.67 | 17.13 |

## 5 CONCLUSION

In this work, we proposed CogVLN, a vision-and-language navigation framework designed for large-scale environments. We introduced the cognitive maps to represent the environment from both visual and semantic perspectives. Building upon this cognitive map, we developed a localization module to determine start and goal positions, a path planning module to generate navigation routes, and a navigation module to execute task while outputting scene information and adjusting routes based on user feedback. Overall, our approach enables interactive and efficient navigation in large-scale environments, demonstrating the effectiveness of cognitive map-guided VLN. In future work, we plan to explore even larger-scale outdoor environments, further improving the efficiency and robustness of the cognitive map.

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

# A APPENDIX

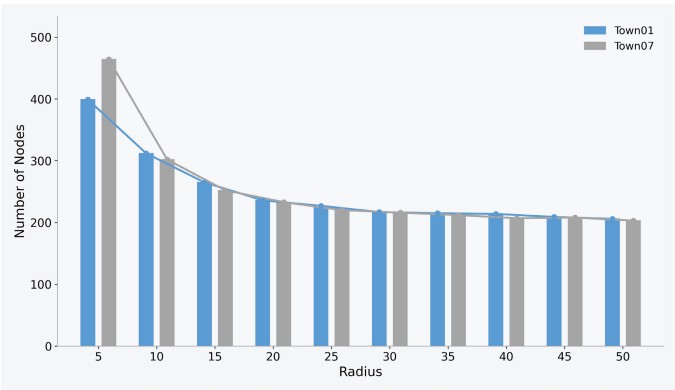

Figure 6: The influence of different radii R on the construction of cognitive maps, (a) to (d) for Town01 and (e) to (g) for Town07.

**The Use of Large Language Models(LLMs)**  We used LLMs solely for language editing (grammar, phrasing, and clarity) and did not use them for study design, method implementation, data analysis, result generation, or technical content writing.

