# OpenReview forum: "CogVLN: Cognitive Map-Guided Vision-and-Language Navigation in Large-Scale Environments"
_ICLR.cc/2026/Conference — ICLR 2026 Conference Withdrawn Submission_

### Official Review · Reviewer_fsfF · 2025-10-25

**Soundness:** 2
**Presentation:** 3
**Contribution:** 2
**Rating:** 4
**Confidence:** 5

**Summary:**

This paper proposes CogVLN, a cognitive map-guided framework for large-scale outdoor Vision-and-Language Navigation (VLN). The core contribution is a "cognitive map" that removes redundant waypoints via visual (DINOv2) and semantic (Grounded-SAM + MLLM + BERT) similarity filtering, retaining key scenes (e.g., intersections). CogVLN also includes three modules (localization, path planning, navigation) that enable interactive navigation via user feedback. Experiments on CARLA Town01/Town07 show CogVLN achieves 34.17%/28.33% success rate (SR) without training, outperforming random policies and equidistant topological maps.

**Strengths:**

1. Cognitive map design: Integrating visual and semantic filtering aligns with human spatial cognition, addressing the trade-off between map compactness and information retention (better than dense point clouds or equidistant maps).

2. Interactive navigation: The user-feedback mechanism effectively solves the long-instruction alignment problem, which is more practical for real-world scenarios (e.g., road blockages) than one-time instruction input.

3. Zero-training utility: Using pre-trained MLLMs and feature encoders avoids tedious environment-specific fine-tuning, lowering the barrier for practical application.

**Weaknesses:**

1. Inadequate baseline comparisons: The paper only compares with random policies and equidistant topological maps, ignoring recent SOTA outdoor VLN methods (e.g., VLN-VIDEO (Li et al., 2024) which uses driving videos for data augmentation, PReP (Zeng et al., 2024) which optimizes MLLM-driven navigation). Without these comparisons, it is impossible to confirm CogVLN’s competitiveness in the field.

2. Limited experimental scope: All experiments are conducted in CARLA simulation, with no real-world testing. Simulated environments lack dynamic noise (e.g., variable lighting, pedestrian occlusion, traffic jams) that exists in real outdoor scenes, making the generalization of CogVLN highly questionable.

3. Unanalyzed key parameters: The cognitive map’s radius $r$ and similarity threshold $\tau\$ directly affect navigation performance, but the paper only shows how $r$ impacts waypoint count (Appendix Figure 6) without explaining why the chosen $r$ is optimal. Also, no parameter sensitivity analysis (e.g., how SR changes with $\tau\$  is provided, reducing the method’s reproducibility.

4. Fragile interaction mechanism: The framework assumes user feedback is always correct, but in practice, users may provide wrong information (e.g., misjudging road blockages). The paper does not propose a mechanism to detect or correct such errors, leading to potential navigation failures in real use.

5. Unassessed real-time performance: Large-scale outdoor navigation requires low latency, but the paper does not report critical metrics like cognitive map construction time or navigation decision latency. It is unclear whether CogVLN can meet real-time requirements.

6. Some related works are missing citations:
[1] NavQ: Learning a Q-Model for Foresighted Vision-and-Language Navigation
[2] Test-time Adaptive Vision-and-Language Navigation
[3] Learning Vision-and-Language Navigation from YouTube Videos
 [4] Magic: Meta-ability guided interactive chain-of-distillation for effective-and-efficient vision-and-language navigation

**Questions:**

See the weakness.

---

> ### Author Response · Authors · 2025-11-29
>
> Thanks you for generous advice making this paper better. They are addressed below.
>
> **W1:** Thank you for your question! The VLN method we propose differs from prior VLN approaches. We have reviewed many previous methods but did not find a suitable baseline for comparison. Whereas prior VLN paradigms mandate strict adherence to step-by-step linguistic instructions, our approach receives only high-level descriptions of the start and goal loci, performs self-localization upon the pre-constructed cognitive map, and subsequently generates a navigation plan. Among them, the PReP (Zeng et al., 2024) method is the most similar to ours; however, since its model weights have not been released, we are unable to conduct a direct comparison.
>
> **W2:** At present, all experiments have been conducted exclusively in the CARLA simulation environment, and real-world validation has not yet been performed. As you correctly pointed out, simulated settings lack many forms of dynamic variability—such as changing lighting conditions, pedestrian occlusions, and traffic congestion—that characterize real-world outdoor scenarios.
>
> Moving from simulation to real-world environments will inevitably expose the system to more complex and less predictable conditions, including the illumination-related challenges you highlighted. To address this, a comprehensive real-world evaluation is planned as an immediate and essential direction for our future work.
>
> **W3:** We will supplement our experiments with an analysis of how the cognitive map radius and similarity threshold affect navigation performance.
>
> As of the present point, the experimental results that have been completed are as follows, the first row of the table represents different radius values.
> - | 8m | 10m | 15m | 20m
> ---|---|---|---|---|
>  Towm01(SR%) | 42.50 | 38.33 | 34.17 | 31.67
>  Town07(SR%) | 40.83 | 34.17 | 28.33 | 26.67
>
> **W4:** We sincerely appreciate your valuable feedback. In our future work, we will incorporate neuro-dynamic error correction and coordinate with human consciousness to make the most accurate decisions. Additionally, we will also consider introducing reinforcement learning, using the feedback from reinforcement learning to help us make correct decisions.
>
> **W5:** We have conducted a systematic evaluation of the real-time performance of our method across its core modules—mapping, localization, and navigation.
>
> During the **cognitive map construction stage**, converting a dense graph with 2,200 vertice into the corresponding sparse cognitive graph (excluding MLLM-based semantic extraction) takes 375 seconds, with an average processing rate of about **5.9 FPS**. In the **navigation stage**, the system runs at an average speed of about **11 FPS**, and the vehicle moves at a real-time speed of approximately **3 m/s**.
>
>
> **W6:** Thank you for your valuable comments! We have cited the corresponding references in the related work section.

---

### Official Review · Reviewer_QWTg · 2025-10-26

**Soundness:** 3
**Presentation:** 3
**Contribution:** 2
**Rating:** 4
**Confidence:** 4

**Summary:**

This paper introduces cognitive map-guided vision-and-language navigation, a method inspired by NeuroBayesSLAM, which aims to improve Vision-and-Language Navigation (VLN) in large-scale environments. The approach proposes a novel cognitive-map-based representation that determines the start and goal locations through visual and linguistic cues and allows dynamic human-agent interaction during navigation.

**Strengths:**

1. Efficient mapping and querying in large-scale environments remain key challenges in VLN. Building upon NeuroBayesSLAM, this work innovatively integrates cognitive mapping principles with visual-language reasoning, partially addressing this long-standing issue.

2. The paper is clearly written, with a well-defined symbol system and few grammatical or formatting errors. Figures are well-designed, minimalistic, and easy to interpret, which enhances readability.

3. The evaluation protocol and metrics are well structured, providing a fair and interpretable framework for measuring performance.

4. The ablation experiments examine several meaningful factors that may influence effectiveness, such as different MLLM backbones and edge pruning within the cognitive map, offering useful insights into the method's robustness.

**Weaknesses:**

1. Runtime and Efficiency Analysis: The paper lacks a quantitative analysis of computational efficiency. It would be beneficial to report runtime comparisons, such as the time cost for start/goal determination and human-agent interaction during navigation.

2. Lines 20–21 and Line 178 claim that humans build spatial representations through cognitive maps.  Please add appropriate theoretical grounding.

3. Line 45: The  paper claims that “building an efficient spatial representation is a prerequisite for navigation”. Please add appropriate theoretical grounding.

4. The experiments would be more convincing if tested on more complex outdoor datasets, such as Touchdown or Map2Seq, which are standard benchmarks for large-scale urban VLN.

5. Real-World Deployment: Please discuss whether the proposed system has been (or can be) deployed in real-world scenarios, and what challenges or limitations are expected in transferring from simulation to reality.

6. Clarify whether the implementation will be open-sourced, which is essential for reproducibility.

7. Figure 1: The framework illustration is visually appealing, but it lacks detailed explanations of how the inputs and outputs of each module interact with subsequent components.

8. Line 86–87: Please add appropriate citations when referring to existing methods.

9. Line 243: The choice of BERT as the encoder needs justification. Why was it selected over other multimodal or VLN-specific language encoders?

10. Line 218: Please explain why DINOv2 was chosen as the visual encoder. How would the performance change if ViT were used?

11. Fix the typo in Line 89. Correct the incorrect Figure reference in Line 213. Resolve reference formatting issues in Line 345. Line 364: The word 'Star' likely should be 'Start'.

**Questions:**

Please see weaknesses.

---

> ### Author Response · Authors · 2025-11-29
>
> Thanks you for generous advice making this paper better. The problems are addressed point-by-point below.
>
>
> **W1:** We will conduct an analysis of the running time and computational efficiency.
>
> Constructing a sparse cognitive graph from a dense graph with 2,200 vertice takes approximately **375 seconds**, corresponding to a processing rate of about **5.9 FPS**.  During navigation, the system runs at around **11 FPS**. For a route consisting of 20 vertice, including both vehicle motion and vertex-level inference, the average time per vertex is approximately **22.8 seconds**.
>
> **W2：** Thank you for your suggestion! We have added the relevant theoretical grounding.
>
> [1] Burgess et al., “The 2014 nobel prize in physiology or medicine: a spatial model for cognitive neuroscience,” Neuron, vol. 84, no. 6, pp. 1120–1125, 2014.
>
> The article describes how humans (and other animals) construct internal spatial representations through a hippocampal–entorhinal cognitive map, implemented by place cells, grid cells, and related neurons, which together provide a neural “GPS” for navigation and episodic memory.
>
> [2] Moser et al., “Place cells, grid cells, and the brain’s spatial representation system,” Annu. Rev. Neurosci., vol. 31, no. 1, pp. 69–89, 2008.
>
> The article provides a comprehensive overview of the brain’s spatial representation system, detailing how place cells, grid cells, and related cell types in the hippocampal–entorhinal circuit interact to form a metric cognitive map that supports flexible navigation and spatial memory.
>
> **W3：** Thank you for your suggestion! We have added the relevant theoretical grounding.
>
> [1] Yu et al. NeuroSLAM: A Brain inspired SLAM System for 3D Environments. Biological Cybernetics, 2019
>
> The article introduces a brain-inspired SLAM system that builds stable 3D cognitive maps of the environment, emphasizing that robust navigation relies on accurate and consistent spatial representations as a foundation.
>
> **W4:** Our methodology diverges fundamentally from conventional Vision-and-Language Navigation (VLN) paradigms, both in terms of data acquisition protocols and utilization schemes. Consequently, conducting experiments on existing benchmarks such as Touchdown and Map2Seq would necessitate a substantial re-engineering effort to restructure their annotation schemas and data pipelines. Owing to temporal constraints, such an undertaking is infeasible within the current project timeline; however, we will accord it high priority in future iterations of this research programme.
>
> **W5:** At present, our system has not been validated outside of simulation; all empirical results reported in this paper were obtained in a controlled virtual environment. We must explicitly acknowledge that deploying the proposed framework in the physical world introduces a non-trivial set of challenges: real-world environments are substantially more stochastic, and robustness will be further stressed by factors such as varying illumination and adverse weather conditions.
>
> **W6:** Once the paper is accepted, we will open-source the entire project.
>
> **W7:** Thank you for your valuable suggestion! In the stage of constructing the cognitive map, the input is **a dense cognitive map**, which includes visual information of vertex and connection information. The output is **a sparse cognitive map**. In the localization and planning stages, the input is **language or visual information** about the star and goal vertex, and the output is **a series of path vertex**. In the navigation stage, the output is the descriptive information along the path. And, we have thoroughly revised Figure 1.
>
> **W8:** Thank you for your suggestion. We have added the relevant literature references.
>
> [1] Tian et al. Loc4plan: Locating before planning for outdoor vision and language navigation. ACM MM,pp 4073–4081,2024.
>
> [2] Vasudevan rt al., Talk2nav: Long-rangevision-and-language navigation with dual attention and spatial memory. IJCV,129(1):246–266,2021.
>
> The above methods all adopt a paradigm in which a complete, long navigation instruction is provided only once: a global language description is given at the beginning of navigation, and throughout execution the agent must continually align its current visual observations with this fixed instruction.
>
> **W9:** Thank you for the suggestion. We chose BERT because its bidirectional contextual modeling captures the full meaning of a word in a single pass, leading to more accurate semantic representations. In our framework, BERT can be replaced by other encoders, including multimodal encoders or language encoders specifically designed for VLN.
>
> **W10:** Thank you for your suggestion. We use DINOv2 because it provides accurate and memory-efficient image features. However, our method is flexible, and DINOv2 can be replaced by other visual encoders, such as ViT or CLIP.
>
> **W11:** Thank you for your suggestion! We have corrected the aforementioned errors and carefully checked the entire manuscript.

---

### Official Review · Reviewer_F1Z3 · 2025-10-27

**Soundness:** 2
**Presentation:** 2
**Contribution:** 2
**Rating:** 2
**Confidence:** 4

**Summary:**

This paper introduces CogVLN, a framework for vision-and-language navigation in large-scale outdoor environments. The authors propose a new method to construct a cognitive map by first generating a dense topological graph and then pruning redundant waypoints using a two-stage process based on visual (DINOv2) and semantic (MLLM+BERT) similarity. This map is then utilized by a modular, zero-shot system composed of localization, planning, and navigation modules, all orchestrated by a MLLM.

A key aspect of the proposed framework is its interactive nature; instead of following a single long instruction, the agent provides descriptions of its current location and adjusts its path based on simple user feedback, allowing for dynamic re-planning.

The system is evaluated in two CARLA environments and claims to achieve excellent performance without any task-specific training.

**Strengths:**

(1) The work addresses the challenging and highly relevant problem of scaling VLN agents to large, complex outdoor environments, which is a critical step toward real-world applications.

(2) I think the core idea of constructing a cognitive map via a two-stage pruning process is a definite strength. The method, which filters redundancy based on both visual appearance and high-level semantics, is well-motivated by principles of human cognition and offers a promising way to create efficient and meaningful environmental representations. So the shift from a traditional one-off instruction-following task to an interactive, human-in-the-loop navigation paradigm is a significant conceptual contribution.

**Weaknesses:**

My main concerns with this submission revolve around the experimental evaluation, which I believe is insufficient to support the paper's claims.

(1).The only baseline presented in Table 1 is a random policy. For a task as complex as navigation, a random agent is expected to have a performance near zero, making it a trivial point of comparison. Without comparisons to any other established or even simple heuristic-based methods, it is impossible to assess whether the reported success rates are actually good.

(2).The paper proposes a novel interactive navigation module. However, its benefit is not isolated. I'd like to know how a non-interactive agent, using the same cognitive map and MLLM-based planner, would perform. Such an ablation would be crucial to demonstrate the true value of the human-in-the-loop component.

**Questions:**

Could you please kindly justify the decision to only compare CogVLN against a random policy? Why were no other baselines, even simple ones, included in the main results table(Table 1) ?

The interactive navigation is a core part of your framework. To understand its contribution, what is the performance of your system if the interaction is removed? For instance, what happens if the agent follows the initially planned path to completion without any user feedback?

And a confusion regarding the task itself: The input seems to be visual and textual descriptions of a start and end point. How does this setup relate to existing outdoor VLN datasets like Touchdown, where the agent is given a series of directions to follow? I'd like to know the rationale behind this particular problem formulation.

---

> ### Author Response · Authors · 2025-11-29
>
> Thanks you for generous advice making this paper better. They are addressed below.
>
> **W1 & Q1:** The VLN method we propose differs from prior VLN approaches. We have reviewed many previous methods but did not find a suitable baseline for comparison. Whereas prior VLN paradigms mandate strict adherence to step-by-step linguistic instructions, our approach receives only high-level descriptions of the start and goal loci, performs self-localization upon the pre-constructed cognitive map, and subsequently generates a navigation plan. Among them, the PReP method is the most similar to ours; however, since its model weights have not been released, we are unable to conduct a direct comparison.
>
> **W2 & Q2：** We sincerely appreciate your question. We must acknowledge our mistake: our wording regarding interactive navigation in the manuscript was inappropriate and led to a misunderstanding. Interactive navigation itself is not the primary focus of this study, and we therefore did not conduct dedicated experiments to evaluate it. The core contribution of our work lies in encoding large-scale environments using a cognitive map representation and enabling navigation through a large language model (LLM); interactive navigation plays only an auxiliary role within this overall framework.
>
> Consequently, none of our experiments involved interactive intervention. In our forthcoming research, we are willing to conduct an additional set of experiments to explicitly quantify the impact of interactivity on navigation performance; owing to time constraints, these results cannot be presented in the current submission.
>
> **Q3：** The input to our task consists of visual and textual descriptions of the start and end points, which differs from existing outdoor VLN approaches. Our design is motivated by the following considerations. First, existing outdoor datasets such as Touchdown are sampled at uniform spatial intervals, leading to both redundancy and gaps in the environmental representation. Second, in current navigation paradigms, the agent typically follows long, step-by-step instructions to determine its actions in different scenes, and this mode of navigation is not well suited to dynamically changing environments.
>
> In contrast, our method constructs a cognitive map to encode scene information. Given only textual or visual descriptions of the start and end locations, the agent can navigate on this map and adapt to dynamic situations during execution, such as temporary road closures.

---

### Official Review · Reviewer_ShMi · 2025-10-28

**Soundness:** 2
**Presentation:** 3
**Contribution:** 2
**Rating:** 2
**Confidence:** 3

**Summary:**

This paper proposes CogVLN, a novel vision-and-language navigation (VLN) framework for large-scale outdoor environments. Inspired by human cognitive mapping, it constructs a compact topological representation of the environment by prioritizing distinctive scenes and filtering redundant ones based on visual and semantic similarity. Leveraging a Multimodal Large Language Model (MLLM), the system features three core modules for localization, path planning, and interactive navigation, where the agent dynamically adjusts its route based on user feedback. Evaluated in CARLA's Town01 and Town07, CogVLN achieves notable success rates (34.17% and 28.33%) without any training, demonstrating robust performance in complex, large-scale settings.

**Strengths:**

1. The cognitive map construction is a key strength, moving beyond dense or equidistant topological maps. By selectively encoding visually and semantically distinct waypoints, it creates a highly compact and efficient representation that aligns with human spatial memory, reducing redundancy and computational load while preserving critical navigational features for large-scale environments.
2. The framework effectively addresses the challenge of aligning long instructions with complex environments by introducing an interactive navigation module. Instead of relying solely on initial instructions, the agent outputs scene descriptions and re-plans routes in real-time based on user feedback, enabling dynamic adaptation and significantly improving robustness in unpredictable scenarios.
3. This paper is good writing.

**Weaknesses:**

1. The paper fails to provide a clear analysis and elaboration of the problem definition. Compare the traditional VLN task (inputs all instructions at once), what is the different and challenge of providing dynamic interactions? Assuming that adding human interaction makes the task easier, a success rate of only 30% is not a very good performance.
2. The paper's empirical validation is significantly weakened by the lack of comparison with established sota methods in indoor or outdoor VLNs. The only benchmark provided is a random policy, which is not a meaningful benchmark. Furthermore, the method lacks deployment in real-world environment to verify its robustness and operational efficiency.
3. The cognitive map construction, while inspired by neuroscience, lacks empirical validation of its superiority over a well-designed equidistant topological map. The ablation shows better performance, but the baseline "Equ" map's construction details are not specified. It is unclear if the performance gain comes from the human interaction or simply from having a more intelligently pruned graph. Additionally, the end-to-end system's computational cost is not discussed. The process involves multiple calls to large models (DINOv2, Grounded-SAM, MLLM) for map construction, localization, and planning, which likely leads to error accumulation and latency, making its applicability to real-time navigation questionable.

**Questions:**

1. The modeling of vision-language navigation (VLN) for drones can be understood as a partially observable Markov decision process that does not rely on known maps for navigation. However, the global map constructed using NeuroBayesSLAM seems to allow the task to navigate on a pre-explored map. Is this inconsistent with the traditional VLN task?
2. The article lacks some details:
    - How is the Equ map in Table 2 constructed?
    - What is the significance of the localization module in locating the starting and target locations?
    - The Navigation Module lacks a detailed description of its specific details. What is the image matching method and the exploration mechanism of the new node?

---

> ### Author Response · Authors · 2025-11-29
>
> Thank you for effort making this paper better. The problems are addressed point-by-point below.
>
> **W1：** （1）**Problem definition:** Our work targets large-scale environments, where we use a cognitive map to effectively represent the scene and combine it with LLMs for vision-language navigation.
>
> （2） Our method is a vision and language method that performs navigation using both visual and language information. Compared with previous VLN tasks, it only requires a language or visual description of the start and goal locations; based on these inputs, it carries out localization and route planning, thereby enabling dynamically adjustable navigation.
>
> （3）Our framework operates in a **zero-shot setting** and does not require any training or fine-tuning to perform navigation tasks in outdoor environments. In addition, the navigation success rate is **affected by both the sparsity of the cognitive map** and the distance between the final reached vertex and the nearest vertex to the goal. In our experiments, we set the topological distance between vertice to 15 meters, while the success criterion is strictly defined as being within 5 meters of the target. As a result, the reported navigation success rate is numerically relatively low, at only about 30%. Experimental results under other topological-distance conditions are presented below.
>
> - | 8m | 10m | 15m | 20m
> ---|---|---|---|---|
>  Towm01(SR%) | 42.50 | 38.33 | 34.17 | 31.67
>  Town07(SR%) | 40.83 | 34.17 | 28.33 | 26.67
>
>
>
> **W2：** (1) **Indirect comparison:** We propose method differs substantially from prior VLN methods. We have reviewed many previous methods but did not find a suitable baseline for comparison. Whereas prior VLN paradigms typically require agents to follow detailed step-by-step linguistic instructions, our approach instead takes only high-level descriptions of the start and goal locations, performs self-localization on a pre-constructed cognitive map, and subsequently generates a navigation plan. Among them, the PReP(zhang et al. 2024) method is the most similar to ours; however, since its model weights have not been released, we are unable to conduct a direct comparison.
>
> (2) **Focus on prototype system:** At the current stage, our work focuses on developing a prototype system that integrates cognitive map construction, localization, and navigation. Moreover, real-world deployment introduces additional challenges, such as heightened environmental dynamics, future work will therefore focus on transferring the proposed methodology from simulated to physical environments.
>
> **W3 & Q2：**（1）**"Equ" experiment setting:** In the “Equ” experiments, we did not introduce any interactive strategy in order to preserve the fairness of the comparison. We construct the “Equ” map by converting the dense map generated by NeuroBayesSLAM into an equidistant topological map. This map does not take into account the visual or semantic content represented by each vertex; it is built purely based on equal spacing.
>
> （2）**Mapping and navigation using LLMs:** Compared to navigation, the mapping stage is more computationally dependent on LLMs: each topological step requires invoking four models, and processing 2,200 vertice takes 375 seconds, corresponding to an average of about **5.9 FPS**.
> The image encodings computed during the mapping stage are cached, so the visual information does not need to be recomputed during navigation. Each decision step requires only a LLM invocation, resulting in a navigation speed of approximately **11 FPS**.
>
> **Q1：** Our approach differs substantially from traditional VLN methods. Traditional VLN systems require a lengthy language instruction as input, obligating the agent to traverse a predetermined route through strict adherence to the specified action sequence. In contrast, our approach conducts navigation upon a pre-explored environmental representation: we employ a cognitive map to encode spatial knowledge, accept only high-level linguistic or visual descriptions of start and goal locations as input, perform localization within this representation, and thereby achieve navigation in large-scale environments
>
> **Q2**
> （2）In our navigation task, the coordinates of the start and goal locations are not directly provided. Instead, the localization module infers them from the input language description or visual image and obtains the coordinates of the  vertice on the cognitive map that are closest to the start and goal.
>
> （3）**Localization is performed using images:** In our method, the input start and goal image information is encoded by the visual encoder, and then a similarity calculation is performed between it and the visual information cached in the  vertice of the cognitive map to determine the specific positions of the start and end points in the cognitive map.
>
> **New  vertice:** Our method handles new  vertice within the cognitive map construction process.

---

### Author Response · Authors · 2025-12-03

We sincerely thank all the reviewers and the area chair for their valuable comments. In this rebuttal, we provide point-by-point responses to each reviewer’s concerns, and the main issues raised by the reviewers are summarized as follows:

- **Lack of Comparison with Existing Methods：** Both our method and prior methods take visual observations and language instructions as inputs and are **all designed for VLN tasks**. The key difference is that our approach first **performs localization on a pre-explored map and then executes navigation**, whereas most existing VLN methods typically **follow a long instruction and navigate step by step based on visual observations**. We conducted a systematic review of previous methods but did not find a suitable baseline for direct comparison. Among them, PReP (Zeng et al., 2024) is the most similar to our method; however, since its model weights have not yet been released, we are unable to perform a direct comparison. We have provided a clear explanation to reviewers ShMi, F1Z3 and fsfF in our response.
- **Use of Large Models & System Runtime Efficiency：** Compared with the navigation phase, the **cognitive map construction** phase is more computationally dependent on large models: it **requires calling four large models**, namely DINOv2, Grounded-SAM, an MLLM, and BERT. In our experiments, processing 2,200 vertice takes 375 seconds, corresponding to an average of about 5.9 FPS. **During navigation**, visual features are retrieved from the cached embeddings generated in the mapping stage and do not need to be recomputed. Therefore, each decision step only **requires a single LLM call**, resulting in a navigation speed of about 11 FPS, with the vehicle moving at approximately 3 m/s. We have already provided reviewers ShMi and fsfF with a detailed explanation regarding the use of large models and computational efficiency.
- **Lack of Experiments in Real-World Environments：** Our method leverages cognitive maps to represent the environment and provides a novel solution for VLN in large-scale environments. At the current stage, our primary focus is on **developing a prototype system** that integrates cognitive map construction, localization, and navigation. All existing experiments are conducted in the CARLA simulator. We acknowledge that real-world deployment will introduce additional challenges, such as dynamic environmental changes and variations in lighting and weather conditions, and addressing these issues will be an important focus of our future work.

We once again thank all reviewers and the area chair for their valuable comments, which have greatly helped us improve this work.

---

### Note · Authors · 2026-01-27

I have read and agree with the venue's withdrawal policy on behalf of myself and my co-authors.

---

### Meta-Review · Area_Chair_R1Jd · 2026-01-05

**Summary:**

This paper presents CogVLN, a new VLN framework that integrates cognitive map construction with MLLM-based planning for large-scale environments.

Reviewers acknowledged the concept of a compact cognitive map and appreciated the modular, zero-shot design. However, all reviewers raised significant concerns regarding experimental validation. Specifically, the work lacks comparison to established VLN methods and real-world testing, which were noted as primary weaknesses.

While the authors clarified that their task (i.e., localization on a pre-built map) differs from standard VLN setups, and noted that the most comparable prior work (PReP) is unavailable for benchmarking, the absence of rigorous comparative evaluation remains a major unresolved issue. Moreover, the interactive navigation module was not experimentally validated in this study.

Although the authors helpfully included computational efficiency metrics (e.g., navigation at ~11 FPS) and corrected an earlier claim regarding interactive navigation, the core evaluation remains insufficient to firmly establish the method's contribution or superiority. The reviewers' overall stance was cautious, with two ratings at the borderline and two recommending rejection, primarily due to these shortcomings.

**Reviewer Concerns:**

See above.

**Reviewer Scores:**

Reviewers may not change their ratings to positive ones.

---

### Decision · Program_Chairs · 2026-01-26

Reject